# Associations between Household Solid Fuel Use, Obesity, and Cardiometabolic Health in China: A Cohort Study from 2011 to 2018

**DOI:** 10.3390/ijerph20042826

**Published:** 2023-02-05

**Authors:** Shihan Zhen, Qian Li, Jian Liao, Bin Zhu, Fengchao Liang

**Affiliations:** 1Shenzhen Key Laboratory of Cardiovascular Health and Precision Medicine, Southern University of Science and Technology, Shenzhen 518055, China; 2School of Public Health and Emergency Management, Southern University of Science and Technology, Shenzhen 518055, China

**Keywords:** solid fuel, obesity, cardiometabolic disease, cohort study

## Abstract

This study aims to explore the longitudinal relationship between solid fuel use and CMD incidence based on a nationally representative follow-up cohort study. A total of 6038 participants of the China Health and Retirement Longitudinal Study (CHARLS) were enrolled in the study. CMD is a cluster of diseases that include heart disease, stroke, and type 2 diabetes. Cox proportional-hazards regression models were used to examine the association between solid fuel use and the incidence or multimorbidity of CMD. The interactions between overweight or obesity and household air pollution on CMD incidence were also investigated. In the present study, solid fuel use from cooking or heating, separately or simultaneously, was positively associated with CMD incidence. Elevated solid fuel use was significantly associated with a higher risk of CMD incidence (HR = 1.25, 95% CI: 1.09, 1.43 for cooking; HR = 1.27, 95% CI: 1.11, 1.45 for heating). A statistically significant interaction between household solid fuel and OW/OB on the incidence of CMD and Cardiometabolic multimorbidity was also observed (*p* < 0.05). Our findings show that household solid fuel is a risk factor for the incidence of CMD. Therefore, reducing household solid fuel use and promoting clean energy may have great public health value for the prevention of CMD.

## 1. Introduction

Cardiometabolic disease (CMD) is a cluster of diseases (including type 2 diabetes (T2D) and atherosclerosis) that can result in cardiovascular disease (CVD), a leading cause of mortality worldwide [1]. The high prevalence of diabetes, cardiovascular disease, and increasing CMD risk factors portend high morbidity and mortality from CMD in China [2]. Meanwhile, it is becoming increasingly common for adults to have several co-occurring diseases [3]. Multimorbidity is associated with a reduced quality of life, higher need for healthcare resources, and greater risk of disability and mortality [4]. Cardiometabolic multimorbidity (CMM), one of the most replicable multimorbidity profiles [5], is defined by the simultaneous coexistence of more than one of the following: coronary heart disease; stroke; and type 2 diabetes [6]. Any combination of these conditions is associated with multiplicative mortality risk, and life expectancy is substantially lower in populations with CMM [7]. The Emerging Risk Factors Collaboration [7] estimated, at age 60, people with cardiometabolic disease (CMD) have a shorter life expectancy of 6–10 years than those with no such disease, while people with CMM have a shorter life expectancy of 15 years. It is meaningful to examine risk factors for the natural history of CMD.

Household air pollution from using solid fuel has caused much concern around the world. The Global Burden of Disease Study (GBD) in 2017 [8] indicated that particulate matter pollution (including household air pollution from solid cooking fuel) was the fourth leading risk factor for deaths in 95 of the 195 countries and territories. Household air pollution from solid fuel use was also a leading risk factor, ranked 3rd in South-East Asia, 6th in East Asia, and 12th in central Asia [9]. Household air pollution from solid fuel mainly refers to exposure to particulate matter with an aerodynamic diameter of less than 2.5 micrometer (PM_2.5_) via the use of polluting fuel (wood, coal, kerosene, charcoal, agricultural residues, or animal dung) for household heating or cooking. It has been listed as an important risk factor for cardiovascular and metabolic diseases [10,11,12]. For example, Yu et al. reported [13] that solid fuel use was associated with higher risks of cardiovascular and all-cause mortality in rural China. Wang et al.’s study [11], using the GBD 2019, estimated that the global deaths and disability-adjusted life years (DALYs) of those with diabetes were attributable to exposures to ambient PM_2.5_, which increased continuously from 1990 to 2019. Globally, 292.5 thousand deaths and 13.0 million DALYs from diabetes were attributed to PM_2.5_ pollution in 2019.

The prevalence of obesity and obesity-related diseases is increasing worldwide. A pooled analysis of 19.2 million participants [14] showed that the global prevalence of obesity doubled from 1975 to 2014. Obesity is linked to a range of chronic disorders, disabilities, and reduced longevity [6,15]. In particular, the increases in the prevalence of overweight or obesity (OW/OB) and abdominal obesity have very strong implications for CMD [6,16,17]. Obesity increases the risk of dyslipidemia and systemic inflammation, which could be common pathways to the development of both diabetes and vascular disease, and an alternative temporal sequence, from obesity to CMD, is also biologically plausible [6]. The updated American Heart Association scientific statement [18] concluded that obese individuals may be a susceptible population at greater risk of cardiovascular events due to PM_2.5_ exposure, and this is a tremendously important public health issue to corroborate. However, the potential impact of OW/OB on the relationship between air pollution and CMD is inconsistent. Several studies have shown that OW/OB may enhance the effects of air pollution on the prevalence of CVDs and stroke [19,20]. However, inconsistent studies were also reported [21,22]. Further, these studies mostly focused on outdoor environment pollution and no epidemiological studies have comprehensively evaluated the role of obesity susceptibility in the relationship between household solid fuel and CMD. Given the paucity of research on these links, the objective of this investigation was to shed light on these relationships.

Thus, this study aims to investigate the association between household solid fuel and the incidence of CMD in older adults using a national wide prospective cohort in China, and further explore the interaction between OW/OB and household solid fuel on CMD incidence.

## 2. Materials and Methods

### 2.1. Study Population

The study was based on the China Health and Retirement Longitudinal Study (CHARLS), an open prospective cohort study. Briefly, the CHARLS is a nationally representative cohort study, covering 450 urban communities and rural areas in 28 provinces of China, which is described in detail elsewhere [23]. For the baseline evaluation, a total of 17,708 participants were recruited from 2011 to 2012 and follow-up assessments were conducted in 2013–2014, 2015–2016, and 2017–2018. The CHARLS was approved by the ethical committees of Peking University. During the investigation, all participants provided written informed consent.

Among the original 17,708 participants, the present study excluded participants with missing data on physical examination (*N* = 4017); household energy source (*N* = 928); age <45 years old at baseline (*N* = 241); the presence of heart disease, stroke, or type 2 diabetes at baseline (*N* = 2314); and missing data during the 7-year follow-up (*N* = 4170). A total of 6038 participants were enrolled in the study. The inclusion and exclusion process of this study’s participants is depicted in Figure 1.

### 2.2. Definition of CMD and CMM

All participants were asked the following: “Have you ever been diagnosed with heart attack, coronary heart disease, angina, congestive heart failure, or other heart problems by a doctor?”; “Have you ever been diagnosed with stroke by a doctor?”; and “Have you ever been diagnosed with diabetes or high hyperglycemia by a doctor?” Participants with affirmative answers were classified as having heart disease, stroke, and diabetes, respectively. Because most participants in our study were aged over 45 years at baseline, it was reasonable to assume that incident cases of unspecified diabetes were mostly T2D. The incident date of CMD was identified as the self-reported date of first diagnosis or the date of blood sampling at the interviews. If two dates were obtained, the earlier one was adopted. In line with a previous study, CMD was defined as the incidence of any one of the positive answers to the above questions [5], and the incident date of CMD was identified as that of the above CMD. CMM was defined as the simultaneous coexistence of more than one positive answer to the above questions (i.e., the coexistence of two or three CMD, including heart disease, stroke, and type 2 diabetes.) [6], and the incident date of CMM was identified as the data of coexistence of two or three above CMD.

### 2.3. Household Air Pollution from Solid Fuel

Household solid fuel in this study was defined and categorized into cooking fuel and heating fuel. Cooking fuel was divided into the following: (1) coal; (2) natural gas; (3) marsh gas; (4) liquefied petroleum gas; (5) electric; (6) crop residue/wood burning; and (7) others. Heating fuel was divided into the following: (1) solar; (2) coal; (3) natural gas; (4) liquefied petroleum gas; (5) electric; (6) crop residue/wood burning; and (7) others. Participants were asked “What is the main source of cooking fuel?” and “What is the main heating energy source?” Among the main sources of cooking fuel and heating energy reported, crop residue/wood burning and coal were defined as solid fuel, while natural gas, marsh gas, liquefied petroleum gas, electricity, solar, and others were defined as clean fuel because they produced significantly less air pollution than solid fuel did. For each participant, the duration of solid fuel use was calculated by the sum of that in all separate time periods (2011–2013, 2013–2015, 2015–2018). The duration of solid fuel use was defined as 0 years if he/her used clean fuel in all four waves. The duration of solid fuel use was defined as more than 7 years if participants used solid fuel in all four waves. For participants who switched the type of fuel used between two points of the interview wave, we estimated the duration of solid fuel use as the midpoint of the two contact dates.

### 2.4. Definition of OW/OB and Abdominal Obesity

Body mass index (BMI) was calculated as weight (kg) divided by the square of the height (m^2^). OW/OB in this study was defined as BMI ≥ 24 kg/m^2^ [24]. Abdominal obesity was assessed using waist circumference (WC) and defined as high if WC ≥ 90 cm for men and ≥80 cm for women [25].

### 2.5. Covariates

The following factors were selected as potential confounding variables: age; gender (female vs. male); marital status (married vs. divorced/separated/widowed/never married); education level (illiterate, primary school, middle school, and high school or above); residence (rural vs. urban); smoking (no vs. yes); and alcohol consumption (no vs. yes).

### 2.6. Statistical Analyses

Baseline characteristics of the participants are presented as percentages by household solid fuel, including both cooking and heating. Based on a longitudinal study with sufficient follow-up periods, the temporal contextual effects of household air pollution on mental health were allowed [26]. The longitudinal relationship between household solid fuel and CMD was explored by Cox proportional-hazards regression models. The endpoint was defined as the incidence or multimorbidity of CMD.

Cox proportional-hazards regression models were conducted to investigate the association between solid fuel use and CMD in the overall population, and a subgroup stratified by OW/OB. Hazard ratios (HRs) with 95% confidence intervals (CIs) were calculated. The models were adjusted for age, gender, education level, marital status, residence, alcohol consumption, and smoking. Results are presented as the HR and 95% CI. Cox proportional-hazards regression models were also employed to examine the interaction between OW/OB and household solid fuel on incidence of CMD. The multiplicative interaction was assessed by introducing a cross-product term in the model. All *p*-values were two-tailed, and differences with *p*-values < 0.05 were considered statistically significant.

## 3. Results

### 3.1. Characteristics of the Participants

The baseline characteristics of the total participant enrolled in the study are presented according to the household fuel types in Table 1. Among the 6038 participants involved in the study, 3108 (51.47%) participants were female, and the average (SD) age of participants was 59.79 (9.47) years. The number of participants with solid fuel exposure from cooking was 3847 (63.71%) and the number of those with exposure from heating was 3776 (62.54%) at baseline. Compared with participants who used clean fuel, participants who used solid fuel tended to be elders, current smokers, and those with a lower education level or living in rural areas.

### 3.2. Association between Household Solid Fuel and CMD or CMM

Table 2 presents the CMD incidence during the cohort associated with household solid fuel. Compared with those using clean fuel, participants using solid fuel showed a statistically significant association with higher CMD incidence (HR = 1.25, 95% CI: 1.09, 1.43 for cooking; HR = 1.27, 95% CI: 1.11, 1.45 for heating). A longer duration of solid fuel exposure from both cooking and heating was positively associated with CMD incidence (HR = 1.06, 95% CI: 1.04, 1.08 for cooking; HR = 1.13, 95% CI: 1.08, 1.17 for heating). Different solid fuel types also showed different effect sizes on CMD incidence. Participants using coal as cooking or heating fuel had a higher incidence of CMD than clean fuel (HR = 1.37, 95% CI: 1.13, 1.67 for cooking; HR = 1.42, 95% CI: 1.22, 1.65 for heating). Associations between household solid fuel and CMD in subgroups stratified by OW/OB are also shown in Table 2. Among participants with OW/OB, solid fuel exposure from heating was found to be positively associated with CMD incidence (HR = 1.38, 95% CI: 1.13, 1.68). Among participants without OW/OB, solid fuel exposure from cooking was found to be positively associated with CMD incidence (HR = 1.42, 95% CI: 1.16, 1.73), and solid fuel exposure from heating was not statistically significant (*p* > 0.05).

Table 3 presents CMM incidence in the cohort associated with household solid fuel. No statistically significant association was observed between solid fuel exposure from cooking and CMM (*p* > 0.05). A longer duration of solid fuel exposure from cooking and heating was positively associated with CMM incidence (HR = 1.08, 95% CI: 1.01, 1.16 for cooking; HR = 1.21, 95% CI: 1.08, 1.34 for heating). Associations between household solid fuel and CMM in subgroups stratified by OW/OB are also shown in Table 3. Among participants with OW/OB, solid fuel exposure from heating was found to be positively associated with CMM incidence (HR = 1.91, 95% CI: 1.09, 3.34), and a longer duration of solid fuel exposure from cooking and heating was also positively associated with CMM incidence (HR = 1.10, 95% CI: 1.01, 1.20 for cooking; HR = 1.26, 95% CI: 1.10, 1.44 for heating). However, among participants with non-OW/OB, no statistically significant association was observed (*p* > 0.05).

The interaction effect between household solid fuel and OW/OB on the incidence of CMD and CMM is shown in Table 4. Participants were divided into four subgroups by fuel usage (cooking or heating) and OW/OB (yes or no). Compared with the participants who used clean fuel for cooking and without OW/OB, the incident risk of CMD and CMM was higher in those using solid fuel and with OW/OB (HR = 2.05, 95% CI: 1.67, 2.51 for CMD; HR = 3.16, 95% CI: 1.76, 5.67 for CMM), those using clean fuel and with OW/OB (HR = 1.78, 95% CI: 1.43, 2.22 for CMD; HR = 2.57, 95% CI: 1.34, 4.96 for CMM), as well as those using solid fuel and without OW/OB (HR = 1.40, 95% CI: 1.15, 1.70 for CMD). Compared with those using clean fuel for heating and without OW/OB, the incident risk of CMD and CMM was higher in those using solid fuel and with OW/OB (HR = 1.93, 95% CI: 1.60, 2.31 for CMD; HR = 3.15, 95% CI: 1.84, 5.39 for CMM), as well as those using clean fuel and with OW/OB (HR = 1.39, 95% CI: 1.12, 1.73 for CMD).

We also assessed the associations between household solid fuel and CMD or CMM incidence in the subgroup stratified by abdominal adiposity. Among participants with abdominal obesity, solid fuel exposure from both cooking and heating was found to be positively associated with CMD incidence. The interaction effect between household solid fuel and abdominal obesity on the incidence of CMD and CMM was statistically significant (Appendix A). To analyze the combined effect of solid fuel for different usages (i.e., cooking and heating), participants were divided into four subgroups (both using clean fuel; cooking with clean fuel while heating with solid fuel; cooking with solid fuel while heating with clean fuel; and both using solid fuel). Compared with the subjects who used both clean fuel for heating and cooking, participants who reported using solid fuel for both tended to present higher CMD and CMM incidence (Appendix A). The effects of fuel type conversion on CMD and CMM incidence are shown in Appendix A. Compared with the participants who persistently used solid fuel for cooking and heating, participants who persistently used clean fuel during the study period showed a lower risk of CMD and CMM incidence. Compared with the participants who persistently used solid fuel for heating, participants who switched heating fuel from solid to clean during the study period showed a lower risk of CMD and CMM incidence (HR = 0.60, 95% CI: 0.46, 0.79 for CMD; HR = 0.47, 95% CI: 0.23, 0.95 for CMM). In addition, after dividing CMD into three specific CMDs (heart disease, stroke, and T2D), we observed a statistically significant association between solid fuel exposure and heart disease and stroke (Appendix A).

## 4. Discussion

This large prospective cohort study presents the notion that solid fuel use is associated with a higher incidence of CMD in a Chinese population. A longer duration of solid fuel use was associated with a higher incidence of CMD. The above associations were lower in those who had switched fuel from solid to clean during the study period. Interactions between household solid fuel and OW/OB in incidence of CMD were statistically significant.

It is becoming increasingly common that adults have several co-occurring diseases [3]; however, evidence on the association between household air pollution and CMM is still limited. Previous studies exploring the implications of household solid fuel use on cardiometabolic risk all focus on cardiometabolic indicators [27,28], while the level of these cardiometabolic indicators cannot accurately represent CMM incidence. Kephart et al.’s [27] cross-sectional study based on 617 participants in Peru assessed the associations between household air pollution and cardiometabolic health. They reported a higher indoor PM_2.5_ was associated with having a 1.15 mmHg systolic BP increase (95% CI: 0.16, 2.86) and a 1.39 mmHg diastolic BP increase (95% CI: 0.52, 2.25), while they did not find associations between indoor air concentrations and CRP or HbA1c outcomes. Because cross-sectional surveys provide weak evidence on the causal association between exposure and outcomes, our longitudinal study with a larger sample size helps to estimate the health effects of long-term exposure to solid fuel use and cardiometabolic health in China more accurately. Another interview study [28] based on several females in rural Honduras assessed the association between cookstove type and health endpoints, and it explored exposure–response associations between kitchen PM_2.5_ and indicators of cardiometabolic health (blood pressure, C-reactive protein, and glycated hemoglobin). They reported that a 25% increase in personal PM_2.5_ was associated with a 10.5% increase in CRP (95% CI: 1.20, 20.6) [29] and a higher BP prevalence (OR = 1.50, 95% CI: 1.00, 2.30) [30]. Different variations and levels of household air pollution, population susceptibility, and region may contribute to the inconsistency in the effects of estimated household solid fuel use on population health. These inconsistences could affect diet habits and physical activity patterns, which may affect the association between household air pollution exposure and CMM incidence. In our study, the proportion of male and female participants were almost equal. They located in both urban and rural areas, and we considered household solid fuel exposure from both cooking and heating. On the basis of studies that examine the association between solid fuel use and cardiometabolic indicators, our longitudinal study helps to add evidence of the interactions between solid fuel use and CMM incidence. Previous studies focusing on the association of household air pollution exposure with separate CMD also partially explained associations with CMM incidence. For example, a meta-analysis [31] demonstrated that the risk of cardiovascular event risk increased by 11% (95% CI: 1.03, 1.19) in those with the highest use of solid fuel versus the lowest use of solid fuel in cohort studies. A cross-sectional study [32] reported no significant association between daily cooking duration and diabetes in the Chinese population. Shin et al. [33] showed a positive association between long-term exposure of ambient air pollutants and CMD in Korean adults. Although the association between solid fuel use and CMD have been suggested in low- and middle-income countries, the magnitude of the association in our study was inconsistent with most studies. This inconsistency may be related to differences in biomass fuel types, population sensitivity, and study designs. The solid fuel sources in our study mainly referred to crop residue/wood and coal, which are different from the studies in developing countries (e.g., dung, wood, or charcoal) [27,31]. The type and levels of gaseous pollutants and PM varies with the source of biomass fuel types [34]. In addition, the present cohort study also estimated the duration of solid fuel use and the incidence of CMD and CMM, and these associations were lower in those who persistently used clean fuel than those who persistently used solid fuel during the study period. Previous studies have reported that switching the type of solid fuel crucially influences the trajectory of related health effects [35,36]. For example, Deng et al. [36]. reported that switching from clean to biomass cooking fuels was associated with an increased risk of hypertension and blood pressure elevations when compared with the persistent use of cleaner cooking fuels. In the present study, we also found that, compared with the participants who persistently used solid fuel for heating, participants who switched from solid to clean fuel for heating showed a lower risk of CMD and CMM incidence. This finding may have important public health significance because switching solid fuel use is a changeable behavior.

Although the underlying mechanisms between household solid fuel use and CMD remain unclear, there are several plausible explanations for this relationship. Low combustion efficiency leads to high levels of various gaseous pollutants and PM from solid fuel burning [34], which are suggested to be positively associated with CMD incidence [11,13,22,37,38]. Studies have shown that PM exposure is significantly associated with several indicators of inflammation [36,39] and oxidative stress [40]. PM can also lead to autonomic nervous system imbalance and resulting hypertension [36]. Household solid fuel use might increase CMD incidence and mortality risk through the above pathways [13,41]. The incomplete burning of solid fuel also produces black carbon, a component of PM, which is also linked to CMD incidence [36,42]. Recent evidence suggested that black carbon may adsorb combustion-derived chemicals (such as complex hydrocarbons and carbonyls, including acetone) that interact with and modify the effect of PM mass on cardiovascular dysfunctions [43,44]. A mixture of PM, NO_2_, SO_2_, CO, and black carbon also yielded a combined effect on cellular mechanisms, including oxidative stress, inflammatory response, atherothrombosis, metabolic abnormalities, decreased endothelial function, and imbalanced autonomic nervous systems [36,45]. Residents usually tend to turn on ventilation when cooking, potentially reducing the levels of pollutants produced by cooking fuel. In addition, dual exposure to solid fuel exposure from cooking and heating may result in overlapping effects and constitute cumulative damage to human health [46]. We discovered a combined effect of solid fuel used for cooking and heating, which may provide evidence of the severe effect of winter on CMD incidence, especially in rural areas.

Our findings support the hypothesis that obesity could modify the impact of household solid fuel use on CMD incidence. Epidemiological evidence suggests that obesity and abdominal fat were positively associated with CMD [47,48]. Obese populations may be more susceptible to the cardiovascular health effects of air pollution for both long-term and short-term exposures [19,49,50,51]. Indeed, among the obese population, antioxidant defense is lower than normal weight participants [48]. Obesity is thought to be a state of chronic oxidative stress and a chronic inflammatory condition [52], and visceral fat accumulation contributes to pro-oxidant and pro-inflammatory states, as well as to alterations in glucose and lipid metabolism [48]. Air pollution is thought to increase oxidative stress [53] and systemic inflammation [54]. Previous studies indicate that the associations between air pollution and increases in inflammatory markers were the strongest in the obese population compared with the normal weight population [51]. Thus, obesity may increase the association between household air pollution on CMD incidence. It is important that obesity may be more susceptible to CMD health effects of air pollution, which could be explained by the fact that the population health impacts of household air pollution increased with the increasing prevalence of obesity, even if ambient concentrations remained stable [19]. If confirmed, robust interactions between obesity and air pollution may provide additional justification in improving the indoor air quality, as well as promoting the maintenance of a healthy bodyweight.

This study’s contribution is the use of a nationally representative database to assess the association between solid fuel use and the incidence of cardiometabolic disease in China. Our findings link CMD risks to solid fuel use. The strengths of this study include the following: a prospective cohort design; multimorbidity profiles; and underscored potential effect of obesity on solid fuel use and CMD incidence. Admittedly, the present study also has several limitations. Firstly, the majority of the population in this study was located in rural regions, and many participants were excluded due to missing data for main variables or being lost during follow-up, which might have led to selection bias. Additional studies are warranted to verify our findings. Because most participants using solid fuel resided in rural regions of China, more measurements for air-quality improvement in these areas are urgently needed. Secondly, solid fuel use was assessed by a self-report questionnaire, rather than by the direct measurement of each individual’s external or internal exposure dose. Due to the lack of data on outdoor pollutants and other chemical exposures, the synergistic effects of multiple pollutants should be considered in future studies. Thirdly, because data availability and some potential confounders, such as socioeconomic and behavior factors, were not considered in the present study, further investigations are needed verify the association between solid fuel exposure and cardiometabolic health in the Chinese population. Finally, because no acute symptoms could be captured to obtain accurate incident dates for non-communicable chronic diseases, the diagnosis date of CMD was treated as the incident date, which might bring some uncertainties due to the possibility that the real incident date of CMD may be earlier. Therefore, the association between solid fuel use and CMD might be underestimated.

## 5. Conclusions

The present study revealed that household solid fuel is an important risk factor for CMD incidence in Chinese adults. Improving fuel structure and reducing the duration of solid fuel use will help decrease the epidemic of CMD in China. Moreover, the interaction effect identified in our study may help to explain the obesity influence on CMD incidence. Understanding that CMD incidence is partially attributable to household air pollution exposure will benefit policy making and health management in CMD prevention because solid fuel use and obesity are potentially modifiable. Further studies on genetic susceptibility and CMD biomarkers in the effect of household air pollution from solid fuel use on CMD are warranted.

## Figures and Tables

**Figure 1 ijerph-20-02826-f001:**
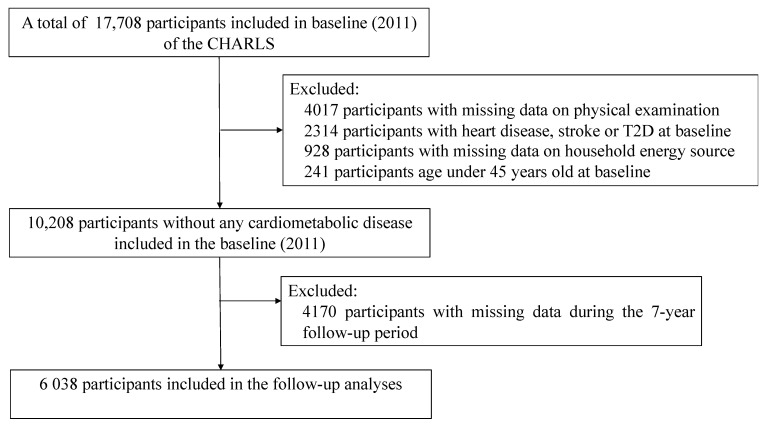
Flow chart of the selection process of participants.

**Table 1 ijerph-20-02826-t001:** Baseline characteristics of 6038 participants by household solid fuel exposure.

Characteristics	Total	Cooking		Heating	
		Clean Fuel	Solid Fuel	Clean Fuel	Solid Fuel
No. n (%)	6038	2191 (36.29)	3847 (63.71)	2262 (37.46)	3776 (62.54)
Age, mean (SD)	59.79 (9.47)	58.36 (9.42)	60.61 (9.40)	59.48 (9.58)	59.98 (9.40)
Gender, n (%)					
Female	3108 (51.47)	1137 (51.89)	1971 (51.23)	1181 (52.21)	1927 (51.03)
Male	2930 (48.53)	1054 (48.11)	1876 (48.77)	1081 (47.79)	1849 (48.97)
OW/OB, n (%)					
No	3903 (64.64)	1308 (59.70)	2595 (67.46)	1437 (63.53)	2466 (65.31)
Yes	2135 (35.36)	883 (40.30)	1252 (32.54)	825 (36.47)	1310 (34.69)
Abdominal adiposity, n (%)					
No	3224 (53.40)	1084 (49.48)	2140 (55.63)	1171 (51.77)	2053 (54.37)
Yes	2814 (46.60)	1107 (50.52)	1707 (44.37)	1091 (48.23)	1723 (45.63)
Marital status, n (%)					
Married	5266 (87.21)	1913 (87.31)	3353 (87.16)	1970 (87.09)	3296 (87.29)
Divorced/separated/widowed/never married	772 (12.79)	278 (12.69)	494 (12.84)	292 (12.91)	480 (12.71)
Education level, n (%)					
Illiterate	1862 (30.84)	489 (22.32)	1373 (35.69)	648 (28.65)	1214 (32.15)
Primary school	2591 (42.91)	949 (43.31)	1642 (42.68)	968 (42.79)	1623 (42.98)
Middle school	1119 (18.53)	495 (22.59)	624 (16.22)	440 (19.45)	679 (17.98)
High school or above	466 (7.72)	258 (11.78)	208 (5.41)	206 (9.11)	260 (6.89)
Residence, n (%)					
Rural	5345 (88.52)	1740 (79.42)	3605 (93.71)	1865 (82.45)	3480 (92.16)
Urban	693 (11.48)	451 (20.58)	242 (6.29)	397 (17.55)	296 (7.84)
Smoking, n (%)					
No	3579 (59.27)	1362 (62.16)	2217 (57.63)	1399 (61.85)	2180 (57.73)
Yes	2459 (40.73)	829 (37.84)	1630 (42.37)	863 (38.15)	1596 (42.27)
Alcohol consumption, n (%)					
No	4010 (66.41)	1435 (65.50)	2575 (66.94)	1498 (66.22)	2512 (66.53)
Yes	2028 (33.59)	756 (34.50)	1272 (33.06)	764 (33.78)	1264 (33.47)

**Table 2 ijerph-20-02826-t002:** HRs (95% CIs) for incident CMD by household solid fuel exposure as stratified by OW/OB.

	CMD								
	Overall (n = 6038)			OW/OB (n = 2135)			Non-OW/OB (n = 3903)		
	Cases	Cases/PYs (1/10,000)	HRs (95%CI)	Cases	Cases/PYs (1/10,000)	HRs (95%CI)	Cases	Cases/PYs (1/10,000)	HRs (95%CI)
Cooking									
Household fuel									
Clean fuel	315	648.01	1.00	173	786.36	1.00	142	533.63	1.00
Solid fuel	652	1341.29	**1.25 (1.09, 1.43)**	268	1218.18	1.13 (0.93, 1.37)	384	1443.07	**1.42 (1.16, 1.73)**
Duration of solid fuel use			**1.06 (1.04, 1.08)**			**1.05 (1.01, 1.08)**			**1.08 (1.04, 1.11)**
Types of household fuel									
Clean fuel	315	648.01	1.00	173	786.36	1.00	142	533.63	1.00
Coal	138	283.89	**1.37 (1.13, 1.67)**	76	345.45	1.23 (0.94, 1.60)	62	232.99	**1.45 (1.08, 1.95)**
Crop residue/wood	514	1057.40	**1.21 (1.05, 1.40)**	192	872.73	1.09 (0.88, 1.34)	322	1210.07	**1.41 (1.15, 1.73)**
Heating									
Household fuel									
Clean fuel	320	658.30	1.00	143	650.00	1.00	177	665.16	1.00
Solid fuel	647	1331.00	**1.27 (1.11, 1.45)**	298	1354.55	**1.38 (1.13, 1.68)**	349	1311.54	1.18 (0.98, 1.42)
Duration of solid fuel use			**1.13 (1.08, 1.17)**			**1.15 (1.09, 1.21)**			**1.11 (1.05, 1.17)**
Types of household fuel									
Clean fuel	320	658.30	1.00	143	650.34	1.00	177	665.16	1.00
Coal	345	709.73	**1.42 (1.22, 1.65)**	187	850.00	**1.49 (1.20, 1.85)**	158	593.76	**1.28 (1.04, 1.58)**
Crop residue/wood	302	621.27	1.12 (0.96, 1.32)	111	504.55	1.22 (0.95, 1.56)	191	717.78	1.10 (0.90, 1.36)

Note: CMD, cardiometabolic disease; HR, hazard ratio; CI, confidence interval; PYs, person years. CMD include heart disease, stroke, and T2D. Multivariable models were adjusted for age, gender, marital status, education level, residence, smoking, and alcohol consumption. *p* values < 0.05 are bold.

**Table 3 ijerph-20-02826-t003:** HRs (95% CIs) for incident CMM by household solid fuel exposure as stratified by OW/OB.

	CMM								
	Overall (n = 6038)			OW/OB (n = 2135)			Non-OW/OB (n = 3903)		
	Cases	Cases/PYs (1/10,000)	HRs (95%CI)	Cases	Cases/PYs (1/10,000)	HRs (95%CI)	Cases	Cases/PYs (1/10,000)	HRs (95%CI)
Cooking									
Household fuel									
Clean fuel	38	576.63	1.00	24	626.63	1.00	14	507.25	1.00
Solid fuel	77	1168.44	1.15 (0.78, 1.69)	43	1122.72	1.14 (0.71, 1.83)	34	1231.88	1.28 (0.66, 2.50)
Duration of solid fuel use			**1.08 (1.01, 1.16)**			**1.10 (1.01, 1.20)**			1.06 (0.96, 1.18)
Types of household fuel use									
Clean fuel	38	576.63	1.00	24	626.63	1.00	14	507.25	1.00
Coal	16	242.79	1.27 (0.71, 2.27)	12	313.32	1.28 (0.65, 2.53)	4	144.93	0.95 (0.30, 2.93)
Crop residue/wood	61	925.64	1.12 (0.74, 1.68)	31	809.40	1.09 (0.65, 1.80)	30	1086.96	1.36 (0.69, 2.71)
Heating									
Household fuel									
Clean fuel	34	515.93	1.00	16	417.75	1.00	18	652.17	1.00
Solid fuel	81	1229.14	1.43 (0.96, 2.13)	51	1331.59	**1.91 (1.09, 3.34)**	30	1086.96	1.00 (0.55, 1.80)
Duration of solid fuel use			**1.21 (1.08, 1.34)**			**1.26 (1.10, 1.44)**			1.13 (0.95, 1.35)
Types of household fuel use									
Clean fuel	34	515.93	1.00	16	417.75	1.00	18	652.17	1.00
Coal	40	606.98	1.50 (0.95, 2.37)	29	757.18	**1.92 (1.03, 3.55)**	11	398.55	0.88 (0.41, 1.89)
Crop residue/wood	41	622.15	1.37 (0.87, 2.15)	22	574.41	**1.90 (1.01, 3.59)**	19	688.41	1.09 (0.57, 2.08)

Note: CMM, cardiometabolic multimorbidity; HR, hazard ratio; CI, confidence interval; PYs, person years. CMD includes heart disease, stroke, and T2D. CMM is defined as occurrence at least two of the above-mentioned diseases. Multivariable models were adjusted for age, gender, marital status, education level, residence, smoking, and alcohol consumption. *p* values < 0.05 are bold.

**Table 4 ijerph-20-02826-t004:** Interactions between household solid fuel exposure and OW/OB on the incident of CMD and CMM.

		CMD			CMM		
Variable	Variable	Cases	Cases/PYs (1/10,000)	HRs (95%CI)	Cases	Cases/PYs (1/10,000)	HRs (95%CI)
Cooking (solid fuel)	OW/OB						
−	−	142	292.12	1.00	14	212.44	1.00
−	+	173	355.89	**1.78 (1.43, 2.22)**	24	364.19	**2.57 (1.34, 4.96)**
+	−	384	789.96	**1.40 (1.15, 1.70)**	34	515.93	1.16 (0.61, 2.18)
+	+	268	551.33	**2.05 (1.67, 2.51)**	43	652.50	**3.16 (1.76, 5.67)**
**Heating (solid fuel)**	OW/OB						
−	−	177	364.12	1.00	18	273.14	1.00
−	+	143	294.18	**1.39 (1.12, 1.73)**	16	242.79	1.58 (0.80, 3.13)
+	−	349	717.96	1.18 (0.99, 1.41)	30	455.24	0.95 (0.53, 1.70)
+	+	298	613.04	**1.93 (1.60, 2.31)**	51	773.90	**3.15 (1.84, 5.39)**

Note: CMD, cardiometabolic disease; CMM, cardiometabolic multimorbidity; HR, hazard ratio; CI, confidence interval; PYs, person years. CMD include heart disease, stroke, and T2D. CMM is defined as occurrence of at least two of the above-mentioned diseases. Multivariable models were adjusted for age, gender, marital status, education level, residence, smoking, and alcohol consumption. *p* values < 0.05 are bold.

## Data Availability

All the data and materials were based on China Health and Retirement Longitudinal Study conducted from 2011 to 2018. Available online: http://charls.pku.edu.cn/ (accessed on 18 January 2023).

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
