# Peer review of "Associations between Household Solid Fuel Use, Obesity, and Cardiometabolic Health in China: A Cohort Study from 2011 to 2018"

_ijerph, 2023, doi:10.3390/ijerph20042826_

Round 1
Reviewer 1 Report
This interesting study aimed to explore the longitudinal relationship between solid fuel use and cardiometabolic disease incidence based on a nationally representative follow-up cohort study in China.
The authors aimed trough a cohort study to determine associations of household solid fuel use, obesity and cardiometabolic health in China.
The present study tried to explore the longitudinal relationship between solid fuel use and cardiometabolic disease incidence based on a nationally representative follow-up cohort study. The results showed that household solid fuel is a risk factor for the incident of cardiometabolic disease. Reducing household solid fuel use and promoting clean energy may therefore have great public health value for the prevention of cardiometabolic diseases.
The authors used the China Health and Retirement Longitudinal Study (CHARLS), a prospective cohort study to collect data in this study. Because CHARLS is a nationally representative cohort study, covering 450 urban communities and rural areas in 28 provinces of China, I consider the sample size to be sufficient for the representativeness to generalise the results. The analysed groups are homogeneous in nationality or their size, gender but not educational lever or residence.
Could it possible to involve more urban population for a higher reporting capacity of the results?
Does the method of heating and cooking with solid fuel concern more of the rural population?
The measurements and instruments used by the authors seem to be valid. The results are processed in detail with statistical confirmation of results.
The discussion is a reasonable extent and includes the essential findings of the study. More literature can be added to the discussion, enriching the authors' arguments.
In view of the limitations of the results reported by the authors, I agree with the authors that for better speaking ability the results of the study, it is needed to additional studies are warranted to verify their findings. It is needed to confirm results which was collected through self-report questionnaire, rather than by the direct measurement of everyone’s external or internal exposure dose. In this study is the lack of data on outdoor pollutants and other chemical exposures, which can have an impact for occurrence cardiometabolic diseases. In the synergistic effects of multiple pollutants should be considered in future studies.
Do the authors plan further research where do they incorporate the missing influencing factors which would provide a broader picture and enable a better understanding in this regard?
The paper I evaluate positively because the study revealed that household solid fuel was an important risk factor for cardiometabolic disease incidence in Chinese adults. Moreover, the interaction effect identified in the study may help to explain the obesity influence in the cardiometabolic disease incidence. Understanding the cardiometabolic disease incidence attributable to household air pollution exposure can help to create benefit policy making and health management in cardiometabolic disease prevention because solid fuel use and obesity are potentially modifiable.
Author Response
Dear Reviewers:
Thank you very much for providing us the opportunity to revise our manuscript entitled “Associations of household solid fuel use, obesity and cardiometabolic health in China: A cohort study from 2011 to 2018” (Manuscript ID: ijerph-2131436). We appreciate the editor and the reviewers for their constructive comments and suggestions, and resubmit it for your consideration.
All the comments from the editors and reviewers were addressed point by point and a detailed list of changes is attached below. The reviewers’ comments are italicized and placed in square brackets. Also, within the revised manuscript, we have used underlined text to highlight changes in response to the reviewers’ comments.
Please see the attachment.
Sincerely,
Fengchao Liang, Ph.D.
Associate Professor
School of Public Health and Emergency Management
Southern University of Science and Technology

Reviewer 2 Report
The manuscript by Shihan Zhen et al. is interesting to the readers by explaining the interactions of overweight (OW) or obesity (OB) with household air pollution on Cardiometabolic disease (CMD) and Cardiometabolic multimorbidity (CMM) incidence. Showing a statistically significant interaction between household solid fuel and OW/OB on the CMD and CMM incidence. Their findings also show that household solid fuel is a risk factor for the incidence of CMD. The manuscript is well-described and well-structured. However, minor corrections are suggested:
1. On Page 7 lines 210-211, the manuscript states that those participants that used solid fuel for cooking and without OW/OB (HR= 1.78, 95% CI: 1.43, 2.22 for CMD; HR= 2.57, 95% CI: 1.34, 4.96 for CMM), in Table 4 this data is showing those that used clean fuel and with OW/OB.
2. On Page 7 line 212, the manuscript states that those participants that used clean fuel for cooking and without OW/OB (HR= 1.40, 95% CI: 1.15, 160 for CMD), in Table 4 this data shows those using solid fuel and without OW/OB.
3. On Page 7 lines 215-216, the manuscript states that those participants that used solid fuel for heating and without OW/OB (HR= 1.39, 95% CI: 1.12, 1.73 for CMD), in Table 4 this data is showing those that used clean fuel and with OW/OB.
4. For Tables S1-S3 an explanation of the results should be included in the manuscript on Page 8.
Author Response
Re: Manuscript No. ijerph-2131436
Title: Associations of household solid fuel use, obesity and cardiometabolic health in China: A cohort study from 2011 to 2018
Dear Reviewer:
Thank you very much for providing us the opportunity to revise our manuscript entitled “Associations of household solid fuel use, obesity and cardiometabolic health in China: A cohort study from 2011 to 2018” (Manuscript ID: ijerph-2131436). We appreciate the editor and the reviewers for their constructive comments and suggestions, and resubmit it for your consideration.
All the comments from the editors and reviewers were addressed point by point and a detailed list of changes is attached below. The reviewers’ comments are italicized and placed in square brackets. Also, within the revised manuscript, we have used underlined text to highlight changes in response to the reviewers’ comments.
Sincerely,
Fengchao Liang, Ph.D.
Associate Professor
School of Public Health and Emergency Management
Southern University of Science and Technology

Reviewer 3 Report
The paper can be accepted for publication.
Reviewer's comments - according to the Table 1 30.84% of the examined individuals are illiterate, while in the limitations on the study it is stated that the examination was carried out by means of a self-reported questionnaire. In the light of this piece of information, how did the illiterate subjects fill in the questionnaire?
Author Response
Re: Manuscript No. ijerph-2131436
Title: Associations of household solid fuel use, obesity and cardiometabolic health in China: A cohort study from 2011 to 2018
Dear Reviewer:
Thank you very much for providing us the opportunity to revise our manuscript entitled “Associations of household solid fuel use, obesity and cardiometabolic health in China: A cohort study from 2011 to 2018” (Manuscript ID: ijerph-2131436). We appreciate your constructive comments and suggestions, and resubmit it for your consideration.
All the comments from the editors and reviewers were addressed point by point and a detailed list of changes is attached below. The reviewers’ comments are italicized and placed in square brackets. Also, within the revised manuscript, we have used underlined text to highlight changes in response to the reviewers’ comments.
Please see the attachment.
Sincerely,
Fengchao Liang, Ph.D.
Associate Professor
School of Public Health and Emergency Management
Southern University of Science and Technology
